# A Tensorized Transformer for Language Modeling

**Xindian Ma**[1], **Peng Zhang**[1]*, **Shuai Zhang**[1],
**Nan Duan**[2], **Yuexian Hou**[1], **Dawei Song**[3], **Ming Zhou**[2]
[1]College of Intelligence and Computing, Tianjin University, Tianjin, China
[2]Microsoft Research Asia, Beijing, China
[3]School of Computer Science and Technology, Beijing Institute of Technology, Beijing, China
{xindianma, pzhang, szhang96, yxhou}@tju.edu.cn
{nanduan, mingzhou}@microsoft.com
{dwsong}@bit.edu.cn

## Abstract

Latest development of neural models has connected the encoder and decoder
through a self-attention mechanism. In particular, Transformer, which is solely
based on self-attention, has led to breakthroughs in Natural Language Processing
(NLP) tasks. However, the multi-head attention mechanism, as a key component
of Transformer, limits the effective deployment of the model to a resource-limited
setting. In this paper, based on the ideas of tensor decomposition and parameters
sharing, we propose a novel self-attention model (namely Multi-linear attention)
with Block-Term Tensor Decomposition (BTD). We test and verify the proposed at-
tention method on three language modeling tasks (i.e., PTB, WikiText-103 and One-
billion) and a neural machine translation task (i.e., WMT-2016 English-German).
Multi-linear attention can not only largely compress the model parameters but also
obtain performance improvements, compared with a number of language modeling
approaches, such as Transformer, Transformer-XL, and Transformer with tensor
train decomposition.

## 1 Introduction

In NLP, Neural language model pre-training has shown to be effective for improving many
tasks [12, 26]. Transformer [35] is based solely on the attention mechanism, and dispensing with
recurrent and convolutional networks entirely. At present, this model has received extensive attentions
and plays an key role in many neural language models, such as BERT [12], GPT [27] and Universal
Transformer [10]. However, in Transformer based model, a lot of model parameters may cause prob-
lems in training and deploying these parameters in a resource-limited setting. Thus, the compression
of large neural pre-training language models has been an essential problem in NLP research.

In literature, there are some compression methods [18, 38, 14] proposed. When the vocabulary is
large, the corresponding weight matrices can be enormous. Tensorized embedding (TE) [18] uses the
tensor-train [25] to compress the embedding layers in Transformer-XL [7], but has not compressed
the attention layer. Recently, Block-Term Tensor Decomposition(BTD) [9] is used to compress
recurrent neural networks (RNNs) [38]. Ye *et al.* [38] propose a compact flexible structure to deal
with the large number of model parameters instead by high dimensional inputs in training recurrent
neural networks (RNNs). This method greatly reduces the parameters of RNNs and improves their
training efficiency. Still, the model only considers the input layer compression by the idea of low-rank
approximation. On the other hand, some methods [14, 2] aim to develop a specific structure on its

weight matrices and can reduce the parameters of the models. However, the new structure after compressing can not be integrated into the model [35].

In Transformer, the multi-head attention is a key part and it is constructed by a large number of parameters. Specifically, Ashish et.al [35] compute the attention function on a set of queries simultaneously, packed together into a matrix $Q$, while the keys and values are also packed together into matrices $K$ and $V$, respectively. The attention function then adopts a no-linear function $softmax$ over two matrices $Q$ and $K$. There are two challenges to find a high-quality compression method to compress the multi-head attention in Transformer.

First, the self-attention function in Transformer is a non-linear function, which makes it difficult to compress. In order to address this challenge, we first prove that the output of the attention function of the self-attention model [35] can be linearly represented by a group of orthonormal base vectors. Then, by initializing a low rank core tensor, we use Tucker-decomposition [33, 20] to reconstruct a new attention representation, where $Q$, $K$ and $V$ can be considered as factor matrices. In order to construct the multi-head mechanism and compress the model, we use the method of Block-Term Tensor Decomposition (BTD), which is a combination of CP decomposition [3] and Tucker decomposition [33]. The difference is that three factor matrices $Q$, $K$ and $V$ are shared in constructing each 3-order block tensor. This process can reduce many parameters.

The second challenge is that the attention model after compressing can not be directly integrated into the encoder and decoder framework of Transformer [35, 7]. In order to address this challenge, there are three steps as follows. First, the average of each block tensor can be computed; Second, multiple matrices can be given by tensor split. Third, the concatenation of these matrices can serve as the input to the next layer network in Transformer. After that, it can be integrated into the encoder and decoder framework of Transformer [35, 7] and trained end-to-end. Moreover, we also prove that the 3-order tensor can reconstruct the scaled dot-product attention in Transformer by a sum on a particular dimension.

Our method combines two ideas which are the low-rank approximation and parameters sharing at the same time. Therefore, it achieves the higher compression ratios. Although the self-attention (i.e., scaled dot-product attention) in Transformer can be reconstructed, we do not consider reconstructing it and choose to split the 3-order tensor (the output of Multi-linear attention) which is helpful for improving the accuracy in experiments.

Our major contributions of this paper are as follows:

1) It is proved that the output of scaled dot-product attention (considering as a function) can be linearly represented by a group of orthonormal base vectors.

2) A novel self-attention method, namely Multi-linear attention, is provided, which combines two compression ideas, parameters sharing and low-rank approximation, together.

3) Multi-linear attention builds the strong connection between three factor matrices (pack a set of queries, keys and values, respectively ), enhancing the ability of capturing sufficient attention information. We also prove our model can reconstruct the scaled dot-product attention in the original Transformer.

In order to validate the benefits of our model, we test it on two NLP tasks, namely language modeling and neural machine translation. In our experiments, the multi-head attention can be replaced by the proposed model, namely multi-linear attention. We have observed that the standard Multi-head attention can be compressed with higher compression ratios on One-Billion dataset. As a result, we show that multi-linear attention not only considerably reduces the number of parameters, but also achieve promising experiments results, especially in language modeling tasks.

## 2 Preliminaries

Multi-linear attention is carried out in this paper. The analysis of Multi-linear attention relies on these concepts and results from the field of tensor decomositon and multi-head attention. We cover below in Section 2.1 basic background on Block-Term tensor decomposition [9]. Then, we describe in Section 2.2 multi-head attention [35].

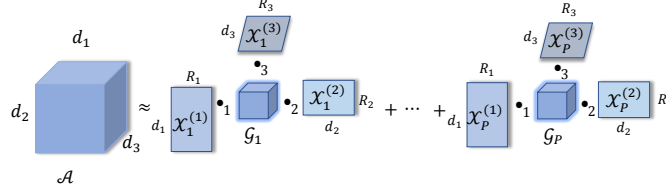

Figure 1: The representation of Block-Term tensor decomposition for a 3-order tensor. $\mathcal{A} \in \mathbb{R}^{d_1 \times d_2 \times d_3}$ is a 3-order tensor, and can be approximated by $P$ Tucker decomposition. $P$ is the CP rank, and $R_1, R_2, R_3$ are the Tucker rank, respectively. In this paper, we assume that $R = R_1 = R_2 = R_3$.

## 2.1 Tensor and Block-Term Tensor Decomposition

**Tensor** We use the Euler script letter $\mathcal{A}$ to denote a tensor which can be thought of as a multi-array. Thereby a vector and a matrix are a 1-order tensor and 2-order tensor, respectively. The element in a $n$-order tensor is denoted as $\mathcal{A}_{d_1,\ldots,d_n}$. In the geometric representation of a tensor, 3-order tensor can be represented by a cube. After that, there is a related concept named *tensor slice* that will be used in this paper. Tensor and some other related concepts are showed in Supplementary Materials A.

**Block-Term Tensor Decomposition (BTD)** Block-Term tensor decomposition is a combination of CP decomposition [3] and Tucker decomposition [33]. Given a $n$-order tensor $\mathcal{A} \in \mathbb{R}^{d_1 \times \ldots \times d_n}$. A high-order tensor can be decomposed into $P$ block terms by the method named BTD. $\bullet_z$ is denoted as the tenor-tensor product on the $z$-th order [19] and $z \in \{1, \ldots, d\}$. Each term contains $\bullet_z$ between a core tensor $\mathcal{G}_i \in \mathbb{R}^{R_1 \times \ldots \times R_d}$ and $d$ factor matrices $\mathcal{X}_i^{(k)} \in \mathbb{R}^{d_k \times R_k}$, where $i \in [1, P]$ and $k \in [1, d]$. The formulation of BTD decomposition is as follows:

$$\mathcal{A} = \sum_{i=1}^{P} \mathcal{G}_i \bullet_1 \mathcal{X}_i^{(1)} \bullet_2 \mathcal{X}_i^2 \bullet_3 \ldots \bullet_d \mathcal{X}_i^{(d)} \tag{1}$$

where $P$ is the CP rank, and $d$ is the Core-order. In our work, the tensor is 3-order. Figure 1 demonstrates the example of how a 3-order tensor $\mathcal{A}$ can be decomposed into $P$ block terms.

## 2.2 Multi-head Attention

In Transformer, the attention function is named as "Scaled Dot-Product Attention". In practice, Transformer [35] processes *query, keys* and *values* as matrices $Q$, $K$, and $V$ respectively. The attention function can be written as follows:

$$Attention(Q, K, V) = softmax(\frac{QK^T}{\sqrt{d}})V \tag{2}$$

where $d$ is the number of columns of $Q$ and $K$. In these work [35, 12, 7], they all use the multi-head attention, as introduced in [35],

$$MultiHeadAttention(Q', K', V') = Concat(head_1, \ldots, head_h)W^O$$
$$where\ head_i = Attention(Q'W_i^Q, K'W_i^K, V'W_i^V) \tag{3}$$

where matrices $W_i^Q$ and $W_i^K \in \mathbb{R}^{d_{model} \times d_k}$, $W_i^V \in \mathbb{R}^{d_{model} \times d_v}$ and $W^O \in \mathbb{R}^{hd \times d_{model}}$. In practice, $d_v$ and $d_k$ are equal to $d$. In this work [35], multiple groups of parameters ($W_i^Q$, $W_i^K$ and $W_i^V$) are used, which results in a large number of redundant parameters.

## 3 Tensorized Transformer

In this section, we first build a Single-block attention in Figure 2 (left) based on the Tucker decomposition, a low-rank decomposition method. In this process, we prove that the self-attention function in Transformer can be represented by a linear function, i.e., a linear combination representation of a set of basic vectors.

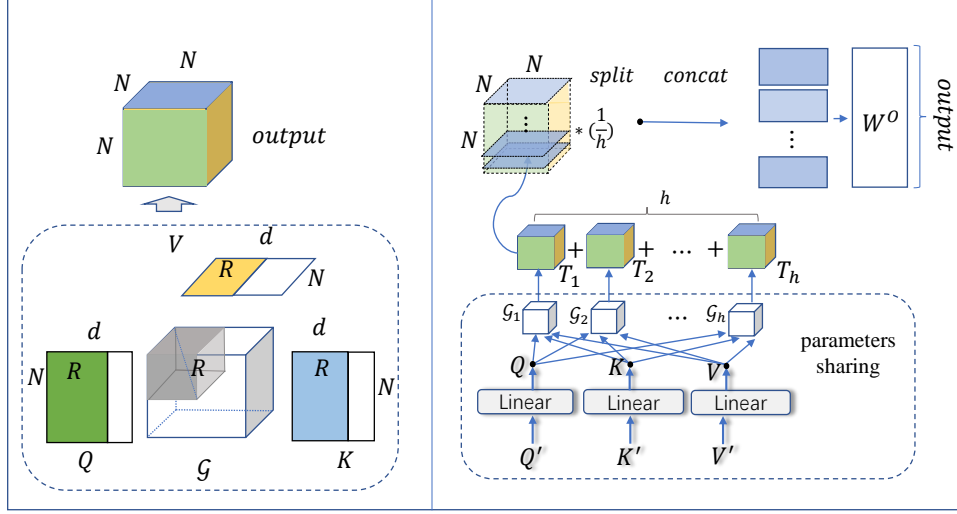

Figure 2: (left) Single-block attention using Tucker decomposition. (right) Multi-linear attention based on Block-Term tensor decomposition.

In order to compress the multi-head mechanism, we propose a multi-linear attention constructed by a Block-Term tensor decomposition. This attention uses the idea of parameters sharing, i.e., sharing factor matrices across multiple blocks, shown in Figure 2 (right). After that, the compression ratios and relatively lower complexity have been analyzed.

## 3.1 Single-block Attention by Tucker Decomposition

Before building the Single-block attention, it is necessary to propose the theorem 3.1. The theorem is closely related to attributes of Single-block attention function by Tucker decomposition [33].

**Theorem 3.1.** *Let $e_1, \ldots, e_n$ be basis vectors from the vector space $S$. Assume that these vectors $e_1, \ldots, e_n$ are linear independent and Q,K,V can be linearly represented by this set of basis vectors. The output of the attention function in Eq. 2 can be represented by a linear combination of the set of these basis vectors.*

$$Attention(Q, K, V) = (e_1, \ldots, e_n)M, \tag{4}$$

*where $M \in \mathbb{R}^{n \times d}$ is a coefficient matrix, and $d$ is a dimension of these matrices (i.e., $Q$, $K$, and $V$).*

*Proof.* The proof can be found in Supplementary Materials B. $\square$

In Figure 2 (left), it is a schematic diagram about the Single-block attention. First, we assume that the query, key and value can be mapped into three factor matrices of which are composed of three groups of orthogonal basis vectors. Three factor matrices are $Q$, $K$ and $V$. After that, we can construct a new attention (i.e., Single-block attention) by initializing a 3-order diagonal tensor (trainable) which is the $\mathcal{G}$. In Figure 2 (left), $R$ is the rank about the tensor, $N$ is the length of a sequence, and $d$ is the dimension of matrix. The function of Single-block attention can be computed based on Tucker-decomposition as follows:

$$
\begin{aligned}
Atten_{TD}(\mathcal{G}; Q, K, V) &= \mathcal{G} \bullet_1 Q \bullet_2 K \bullet_3 V \\
&= \sum_{i=1}^{I} \sum_{j=1}^{J} \sum_{m=1}^{M} \mathcal{G}_{ijm} Q_i \circ K_j \circ V_m
\end{aligned}
\tag{5}
$$

where $\mathcal{G}$ is a core tensor. $i, j$ and $m$ are the indexes of the core tensor. $\circ$ is the outer product. $\bullet_z$ is the same definition in Eq. 1. $Q_i, K_j$ and $V_k$ are column vectors from matrices $Q$, $K$ and $V$, where $Q \in \mathbb{R}^{N \times d}$, $K \in \mathbb{R}^{N \times d}$ and $V \in \mathbb{R}^{N \times d}$, and $N$ is the length of a sequence. In practice, we set $I=J=M=R$. The core tensor $\mathcal{G}$ can be defined as follows,

$$
\mathcal{G}_{ijm} = \begin{cases} rand(0,1) & i = j = m \\ 0 & otherwise \end{cases}
\tag{6}
$$

where the $rand(0, 1)$ is a random function, and the diagonal entries of core tensor $\mathcal{G}$ form the vector $\boldsymbol{g}$. Each entry $\boldsymbol{g}_r \in (0, 1)$, $r \in \{1, \ldots, R\}$. We can consider $\boldsymbol{g}$ as the trainable weight. In experiments, we compute the weight vector by $softmax$ function (i.e., $softmax(\boldsymbol{g})$).

After that, the output of Single-block attention function is a 3-order tensor which is given by linear computation. The Single-block attention (i.e., a 3-order tensor with Tucker decomposition) can reconstruct the *Scaled Dot-Product attention* in Eq. 2 by the summing over the tensor according to the second index [2] (it can be seen as the coordinates in the vertical direction for a tensor), as proved in the following corollary. Note that in our model, we do not adopt the above reconstructing process. Instead, to obtain a new representation, we adopt the concat method after the tensor splitting (see Sec. 3.2). We will further show the compression ability of the Single-block attention in Sec. 3.3.

**Corollary 1.** *Under the same conditions as in Theorem 3.1 and the value of $N$ is equal to the value of d, Single-block attention representation Eq. 5 can reconstruct the Scaled Dot-Product attention in Eq. 2 by the summing over the tensor (i.e., the output of Single-block attention function) according to the second index. It holds that:*

$$Attention(Q, K, V)_{i,m} = \sum_{j=1}^{N} Atten_{TD}(\mathcal{G}; Q, K, V)_{i,j,m} \qquad (7)$$

*where $i$, $j$ and $m$ are the indices of the Single-block attention's output (i.e., a 3-order tensor). $Atten_{TD}(\cdot)$ is the function of Single-block attention based on Tucker decomposition. $i$ and $m$ are the indices of outputs (i.e., a matrix) from Eq. 2.*

*Proof.* The proof can be found in Supplementary Materials C. ☐

## 3.2   Multi-Linear Attention by Block-Term Tensor Decomposition

In order to construct the multi-head mechanism and compress the parameters of multiple groups of mapping, we use a group of linear projections, and share the output from the linear projections. In Figure 2(right), the learned linear projection can map queries, keys and values to three matrices which are composed of basis vectors. After that, we use the Block-Term tensor decomposition to build multi-head mechanism. In our work, our model is named as Multi-linear attention, which can be formulated as follows:

$$MultiLinear(\mathcal{G}; Q', K', V') = SplitConcat(\frac{1}{h} * (T_1 + \ldots + T_h))W^O$$
$$where \ T_j = Atten_{TD}(\mathcal{G}_j; Q'W^q, K'W^k, V'W^v) \qquad (8)$$

where the core tensor $\mathcal{G}_j$ is a diagonal tensor, and the number of parameter in $\mathcal{G}_j$ is equal to the rank of core tensor, $j \in \{1, \ldots, h\}$. $Q'W^q$, $K'W^k$ and $V'W^v$ are equal to $Q$, $K$ and $V$ in Eq. 5, respectively. $\mathcal{G}$ is the set of the core tensors. $SplitConcat(\cdot)$ is a function which achieves the concatenation after splitting for a 3-order tensor. Figure 2 (right) shows the basis idea about the multi-linear attention. The $W^O$ is the parameter matrix which is a full connection layer and correlated to the output of Multi-linear attention. $Atten_{TD}(\cdot)$ is the function of Single-block attention, which is a part of Multi-linear attention. $W^q$, $W^k$ and $W^v$ are the parameters matrices which are shared in constructing Multi-linear attention.

The Multi-linear attention is a compression model. After compressing the multi-head attention in Transformer, it is to achieve a Tensorized Transformer. The Multi-linear attention can be incorporated into Transformer architecture. A diagram which is about the incorporating of Multi-linear attention in partial Transformer structure is given in Supplementary Materials E.1.

## 3.3   Analysis of Compression and Complexity

**Compression** Our focus is on the compression of the multi-head mechanism in the multi-head attention of Transformer. Previous work [35] gets the multi-head attention by multiple groups of linear mappings. We use three linear mappings for matrices $Q$, $K$ and $V$, respectively. For the output of three mappings, we choose to share them which are considered as three factor matrices in

reconstructing the Multi-linear attention. This process is shown in Figure 2 (left). $h$ is the number of heads in [35], and $d$ is the dimension of factor matrices. The compression ratios can be computed by $(3 \times h \times d)/(3 \times d + h)$. In practice, $h$ is normally set to 8, $d$ is set to 512. In this case, the compression ratios can achieve 8. In other words, we can reduce almost 8 times parameters in the attention layer. The details of the computing of compression ratios can be found in Supplementary Materials D. The Transformer also contains other network layers, such as Position-wise feed forward network and embedding layers et al. Therefore, for the compression ratios in whole Transformer, we can compare it by the analysis of experimental results for model parameters.

**Complexity** The time complexity of the attention function in Eq. 2 is $\mathcal{O}(N^2 d)$, $N$ is the length of a sequence, and $d$ is the representation dimension. In Multi-linear attention, we can reorder the computations to receive the model complexity $\mathcal{O}(N^3)$, where $N$ is also the length of the sequence. The minimum number of sequential operations in Multi-linear attention for different layers is approximately equal to the self-attention in Transformer [35].

# 4    Related Work

The field of language modeling has witnessed many significant advances. Different from the architectures of convolutional neural network (CNNs) and recurrent neural networks (RNNs) language modeling, the Transformer [35] and its variants [7, 12, 10] achieve excellent results in language modeling processing. Transformer networks have a potential of learning long-term dependency, but are limited by a fixed-length context in the setting of language modeling. Vaswani *et al.* [35] uses a segment-level recurrence mechanism and a novel positional encoding scheme to resolve this question. BERT [12] is a kind of bidirectional encoder representations from transformers. It is designed to pre-train deep bidirectional representation and obtains new SoTA on some NLP tasks. Although these methods have achieved great results, a large number of parameters make it difficult for the model to be trained in limited resources. Transformer fails to generalize in many simple tasks, e.g. copying string and logical inference [10]. Universal Transformers [10] propose a self-attentive recurrent sequence model which addresses this problem. This methods can increase the training speed. In their work, authors following weight sharing found in CNNs and RNNs, extend the Transformer with a simple form of weight sharing that strikes an effective balance between induces and model expressivity. This methods also uses a large number of parameters.

Therefore, it is very important to consider how to reduce the amount of memory and computing they need. As we know, existing model compression methods are mainly divided into parameter pruning and sharing [14], low rank approximation [29], knowledge transfer [2], and transferred convolutional filters [6]. Currently, tensor decomposition methods are used to decompose a high-order tensor, which can get different neural network language model structures [39, 40]. Besides, tensor decomposition methods which adopts the idea of low rank approximation in most cases, have been successfully applied to neural networks compression. For example, in literature [11, 16], researchers approximate a tensor by minimizing the reconstruction error of the original parameters on convolutional neural networks (CNNs). However, these approaches tend to accumulate errors when multiple layers are compressed sequentially, and the output feature maps deviate far from the original values with the increase of compressed layers. Our compression method uses the idea of parameters sharing in the constructing of attention layers, and the size of output is same as the output from self-attention in Transformer which can effectively avoid these problems. Tensorizing Neural Networks [24] have combined the idea of reshaping weights of fully-connected layers into high-dimensional tensors and representing them in Tensor Train format [25]. This approach was later extended to convolutional [13] and recurrent neural networks [36]. Recently, in these work [5, 34], researchers introduce efficient compression methods for the embedding and $softmax$ layers based on structured low rank matrix approximation. TT-embedding [18] aims to compression the larger embedding layer on Transformer-XL [7]. Sparse Transformer [28] adopts sparse techniques on the attention matrix and reduces its parameters. This work uses a sparse attention matrix by selecting the information on some positions in the attention matrix, but does not change the mechanism of the attention. Our method is different from these works, and combines two compression idea (low rank approximate and parameters sharing) to construct a tensorized Transformer.

In our work, we focus on the compression the multi-head attention in Transformer based the idea of parameters sharing. At the same time, we also combine low-rank approximate method to reduce parameters and computation complexity.

# 5 Experiments

Transformer is a versatile and powerful modeling tool and widely is used in various natural language process tasks. In order to verify the effectiveness of our method (i.e., Multi-linear attention) replacing multi-head attention in Transformer, we carry out two NLP tasks named language modeling (LM) and neural machine translation (NMT). Code[3] for running experiments has been released, and the key code which is about our method can be found in Supplementary Materials F.

## 5.1 Language Modeling

Language modeling is the task of predicting the next word in a sentence. This task is to estimate the joint probability $p(s)$ of a sentence of tokens $s=(w_1, \ldots, w_n)$. The resulting models can be used to generate text or further fine-tuned to solve other NLP tasks [27]. In this paper, we employ the standard setting of predicting next token given the sequence of preceding tokens, based on the function $p(s) = p(w_1) \prod_{i=2}^{n} p(w_i|w_1, \ldots, w_{i-1})$. We chose three datasets in the order of small (i.e., PTB), medium (i.e., WikiText-103) and large (i.e., One-Billion). Models are evaluated based on Perplexity (PPL), which is the average per-word log-probability. The lower the PPL, the better the model is.

Specially, we take Transformer, the open source state-of-the art language modeling architecture, and replace the standard multi-head attention layers with our Multi-linear attention. Then, we test different model configurations on the PTB [23], WikiText-103 [22] and One-Billion Word benchmark [4] datasets and report the results in Table 1 and Table 2.

Table 1: Results (PPL) and model parameters with state-of-the-art results on One-Billion. Tensorized Transformer is our model. The core-1 is that the model use Single-block term tensor. Analogously, the core-2 is that two block term tensor is used.

| Model | Params | Test PPL |
|---|---|---|
| RNN-1024+9 Gram [4] | 20B | 51.3 |
| LSTM-2018-512 [17] | 0.83B | 43.7 |
| GCNN-14 bottleneck [8] | – | 31.9 |
| LSTM-8192-1024+CNN Input [17] | 1.04B | 30.0 |
| High-Budget MoE [32] | 5B | 28.0 |
| LSTM+Mos [37] | 113M | 37.10 |
| Transformer+adaptive input [1] | 0.46B | 23.7 |
| Transformer-XL Base [7] | 0.46B | 23.5 |
| Transformer-XL Large [7] | 0.8B | 21.8 |
| Tensorized Transformer core-1 | 0.16B | 20.5 |
| Tensorized Transformer core-2 | 0.16B | **19.5** |

## 5.2 Results and Details

PTB has $929k$ training tokens, $73k$ validation words, and $82k$ test words. The results is reported in Table 2. Similar to AWD-LSTM-MoS [37], we apply variational dropout and weight average to our model (i.e., Tensorized Transformer). In addition, we need to state that, our model only replaces the multi-head attention using Multi-linear attention structure, and the other structures remain the same. We compare the results of our model with other models. Our model achieves the comparable results with SoTA when the number of core tensor is equal to two. However, our model size (i.e, model parameters) reduces by nearly half comparing with Transformer and Transformer-XL.

WikiText-103 contains 267,735 unique tokens. The dataset is available word-level language modeling benchmark with long-term dependency. It contains 103M training tokens from $28k$ articles, with an average length of 3.6k tokens per article, which allows testing the ability of long-term dependency modeling. As shown in Table 2, our model get the perplexity of $18.9$, which is a comparable experimental result with the previous SoTA perplexity $18.3$ , which demonstrates the effectiveness of the proposed attention architecture.

| Model | PTB | | | WikiText-103 | | |
|---|---|---|---|---|---|---|
| | Params | Val PPL | Test PPL | Params | Val PPL | Test PPL |
| LSTM+augmented loss [15] | 24M | 75.7 | 48.7 | – | – | 48.7 |
| Variational RHN [41] | 23M | 67.9 | 65.4 | – | – | 45.2 |
| 4-layer QRNN [21] | – | – | – | 151M | – | 33.0 |
| AWD-LSTM-MoS [37] | 22M | 58.08 | 55.97 | – | 29.0 | 29.2 |
| Transformer+adaptive input [1] | 24M | 59.1 | 57 | 247M | 19.8 | 20.5 |
| Transformer-XL-Base [7] | 24M | 56.72 | 54.52 | 151M | 23.1 | 24.0 |
| Transformer-XL-Large [7] | – | – | – | 257M | – | **18.3** |
| Transformer-XL+TT [18] | 18 M | 57.9* | 55.4* | 130M | 23.61* | 25.70* |
| Sparse Transformer [28] | 14M | 74.0* | 73.1* | 174M | 38.98* | 40.23* |
| Tensorized Transformer core-1 | 12M | 60.5 | 57.9 | 85.3M | 22.7 | 20.9 |
| Tensorized Transformer core-2 | 12M | **54.25** | **49.8** | 85.3M | **19.7** | 18.9 |

Table 2: Results and compression with state-of-the-art results on PTB and WikiText-103. '−' indicates no reported results in that setting, '*' indicates that the results is our own implementation.

The One-Billion Word benchmark is a large dataset derived from a news site. The dataset consists of $829,250,940$ tokens over a vocabulary of $793,471$ words. In this dataset, sentences are shuffled and hence the context is limited. Consequently, this dataset mainly tests the ability of modeling only short-term dependency. The comparison between Tensorized Transformer and the other methods are shown in Table 1. Although Tensorized Transformer is mainly designed to better compress Transformer or Transformer-XL model, it dramatically improves the single-model SoTA from **21.8** to **19.5**. Specifically, Tensorized Transformer significantly outperforms a contemporary method using vanilla Transformers [35], suggesting that the advantage of the tensorized Transformer is also generalizable to modeling short sequences.

Table 2 and Table 1 show that our model get the lower PPL than other models in three datasets. An exciting observation is that our model has much fewer parameters. The model of Transformer-XL+TT [18] is a recent compression model with Tensor Train to compress the input embedding layers only. Sparse Transformer [28] uses the method of sparse attention matrix to compress Transformer model. The results in Table 2 show that compared with Transformer-XL+TT, our method has much fewer parameters, and better language modeling performance. These results verify that our model (i.e., Multi-linear attention) is effective in language modeling tasks, and has performed well for the model compression. Other details (such as hyperparameters and Hardware) can be found in Supplementary Materials E.

## 5.3 Neural Machine Translation

The goal is to map an input sequence $s = (x_1, x_2, \ldots, x_n)$ representing a phrase in one language, to an output sequence $y = (y_1, y_2, \ldots, y_m)$ representing the same phrase in a different language. In this task, we have trained the Transformer model [35] on WMT 2016 English-German dataset [31]. Sentences were tokenized using the SentencePiece [4]. For our experiments, we have replaced each of the attention layers with Multi-linear attention in Encoder. For evaluation we used beam search with a beam size of 5 and length penalty $\alpha$=0.6. In this section, we only compared the results with Transformer [35]. Our results are summarized in Table 3. $*$ indicates that the result is our own implementation.

In Table 3, we select two baseline models. The Base-line [31] is first model in WMT 2016 English-German dataset. For the other baseline, we use the basic Transformer architecture [35]. The BLEU score is $34.5$ for the basic architecture. We carry out two Tensorized Transformer structures, namely core-1 and core-2 respectively. When Tensorized Transformer core-1 and core-2 are used, the BLEU scores are $34.10$ and $34.91$, which achieves better performance over Transformer. As for the reported model parameter size, our model uses less parameters.

Table 3: Results and compression with Transformer on WMT-16 English-to-German translation.

| Model | Params | BLEU |
|---|---|---|
| Base-line [31] | – | 26.8 |
| Linguistic Input Featurec [30] | – | 28.4 |
| Attentional encoder-decoder + BPE [31] | – | 34.2 |
| Transformer [35] | 52M | 34.5* |
| Tensorized Transformer core-1 | 21M | 34.10 |
| Tensorized Transformer core-2 | 21.2M | **34.91** |

## 5.4 Discussion

We have shown the results on language modeling and neural machine translation tasks using the Multi-linear attention. For the compression of the model parameters, although we report the parameters of the whole model structure, our method mainly considers the compression of multi-head attention but has not changed other layers in Transformer. Regarding the rationale for the improvements, in Corollary 1, we prove that the output of the original attention can be represented by summing over the 3-order tensor. In Figure 2, we use a concat function over these matrices from tensor splitting. The operation of concat can model all values in the 3-order tensor, and thus captures more information than sum operator. Another reason could be the alleviation of overfitting by reducing parameters. The overfitting will appear when the number of the core tensor is greater than 2. Besides, according to our experiments, relatively large dimensions of the word embedding can lead to overfitting, resulting in performance degradation. Therefore, our model requires a relatively small dimension of the embedding, compared with the original Transformer. In order for a more systematic evaluation, we report more experiments and analyses in Supplementary Materials E.4.

## 6 Conclusion and Further Work

We have proposed a novel self attention encoder layer, namely the Multi-linear attention, to compress the original multi-head attention and derive a novel encoding scheme. Our main contribution lies in a structure of Tensorized Transformer based on Block-Term tensor decomposition which is represented by the combination of a group of 3-order tensors, with low-rank approximation and parameters sharing ideas adopted. Compared with existing Transformer based methods, our model achieved higher compression ratio and got better experimental results, particularly in language modeling task. These evidences imply that our method can potentially be further applied to more NLP tasks with limited resources.

In the future, we will continue to optimize the Tensorized Transformer framework and apply it in other NLP tasks. As we stated earlier, our model may suffer from overfitting when the number of cores is large in language modeling. In the future, we will explore the fundamental reasons that cause the problem and tackle them within the Tensorized Transformer framework.

## 7 Acknowledgement

This work is supported in part by the state key development program of China (grant No. 2017YFE0111900, 2018YFC0831704), Natural Science Foundation of China (grant No. 61772363, U1636203), and the European Unions Horizon 2020 research and innovation programme under the Marie SkodowskaCurie grant agreement No.721321.

## Footnotes

*Corresponding Author: Peng Zhang

[2]If the coordinates of a 3-order tensor are $i$, $j$ and $m$, $j$ is the second index.

[3]https://github.com/szhangtju/The-compression-of-Transformer

[4]https://github.com/google/sentencepiece

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
