[Supplementary Material]

# Supplementary Material for A Tensorized Transformer for Language Modeling

**Xindian Ma**[1], **Peng Zhang**[1]*, **Shuai Zhang**[1],
**Nan Duan**[2], **Yuexian Hou**[1], **Dawei Song**[3], **Ming Zhou**[2]
[1]College of Intelligence and Computing, Tianjin University, Tianjin, China
[2]Microsoft Research Asia, Beijing, China
[3]School of Computer Science and Technology, Beijing Institute of Technology, Beijing, China
{xindianma, pzhang, szhang96, yxhou}@tju.edu.cn
{nanduan, mingzhou}@microsoft.com
{dwsong}@bit.edu.cn

## A   Tensor and Tensor Slice

As introduced in [2], a tensor and the tensor slice can be defined as follows.

**Definition 1** (tensor). *Let $D_1$, $D_2$, ..., $D_N \in \mathbb{N}$ denote index upper bounds. A tensor $\mathcal{A} \in \mathbb{R}^{D_1,...,D_N}$ of order $N$ is an $N$-way array where elements $\mathcal{A}_{d_1,d_2,...,d_n}$ are indexed by $d_n \in \{1, 2, ..., D_n\}$ for $1 \leq n \leq N$.*

The concept of tensor slice is specified as:

**Definition 2** (tensor slice). *A tensor slice is a two-dimensional section (fragment) of a tensor, obtained by fixing all indexes except for two indexes.*

## B   Theorem 3.1

Let $e_1, \ldots, e_n$ be basis vectors from the vector space $S$. Assume that these vectors $e_1, \ldots, e_n$ are linear independent and $Q,K,V$ can be linearly represented by this set of basis vectors. The output of self-attention function in Eq. 2 (in the paper) can be represented by a linear combination of the set of these basis vectors.

$$Attention(Q, K, V) = (e_1, \ldots, e_n)M, \qquad (1)$$

where $M \in \mathbb{R}^{n \times d}$ is a coefficient matrix, and $d$ is a dimension of these matrices (i.e., $Q$, $K$, and $V$).

*Proof.* If $Q$, $K$ and $V \in \text{Span}(e_1, \ldots, e_n)$, the linear combination representation of matrices $Q,K$ and $V$ can be written as follows:

$$\begin{cases} Q = (e_1, e_2, \ldots, e_n)(\alpha_1, \alpha_2, \ldots, \alpha_d) \\ K = (e_1, e_2, \ldots, e_n)(\beta_1, \beta_2, \ldots, \beta_d) \\ V = (e_1, e_2, \ldots, e_n)(\xi_1, \xi_2, \ldots, \xi_d) \end{cases} \qquad (2)$$

The self-attention function is written as follows [8]:

$$Attention(Q, K, V) = softmax(\frac{QK^T}{\sqrt{d}})V, \qquad (3)$$

where $QK^T$ can be computed as follows:

$$QK^T = (\boldsymbol{e}_1, \boldsymbol{e}_2, \ldots, \boldsymbol{e}_n)(\boldsymbol{\alpha}_1, \boldsymbol{\alpha}_2, \ldots, \boldsymbol{\alpha}_d)(\boldsymbol{\beta}_1, \boldsymbol{\beta}_2, \ldots, \boldsymbol{\beta}_d)^T (\boldsymbol{e}_1, \boldsymbol{e}_2, \ldots, \boldsymbol{e}_n)^T \quad (4)$$

As a result, the input of $softmax$ function is a product of coefficient matrices $(\boldsymbol{\alpha}_1, \ldots, \boldsymbol{\alpha}_d)$ and $(\boldsymbol{\beta}_1, \ldots, \boldsymbol{\beta}_d)^T$. Then, we have

$$softmax(\frac{QK^T}{\sqrt{d}}) = (\boldsymbol{e}_1, \ldots, \boldsymbol{e}_n)softmax(A/\sqrt{d})(\boldsymbol{e}_1, \ldots, \boldsymbol{e}_n)^T \quad (5)$$

where the matrix $A$ is equal to $(\boldsymbol{\alpha}_1, \ldots, \boldsymbol{\alpha}_d)(\boldsymbol{\beta}_1, \ldots, \boldsymbol{\beta}_d)^T$. Therefore, the attention representation can be written as follows:

$$softmax(\frac{QK^T}{\sqrt{d}})V = (\boldsymbol{e}_1, \boldsymbol{e}_2, \ldots, \boldsymbol{e}_n)\, softmax(A/\sqrt{d})(\boldsymbol{\xi}_1, \boldsymbol{\xi}_2, \ldots, \boldsymbol{\xi}_d)$$
$$= (\boldsymbol{e}_1, \boldsymbol{e}_2, \ldots, \boldsymbol{e}_n)\, M \quad (6)$$

where the matrix $M$ is equal to $softmax(A/\sqrt{d})(\boldsymbol{\xi}_1, \boldsymbol{\xi}_2, \ldots, \boldsymbol{\xi}_d)$. The $softmax(A/\sqrt{d})$ is to normalize the coefficient matrices of $Q$ and $K$. It turns out that the output of the attention function [8] can be represented by a linear combination of the set of basic vectors. $\square$

After the proof, it is helpful to describe the basic idea. First, we consider that the self-attention function can be linearly represented by a set of orthogonal basis vectors, when the input of $softmax$ function is the product of two coefficient matrices, $(\boldsymbol{\alpha}_1, \boldsymbol{\alpha}_2, \ldots, \boldsymbol{\alpha}_d)$ and $(\boldsymbol{\beta}_1, \boldsymbol{\beta}_2, \ldots, \boldsymbol{\beta}_d)^T$, respectively. Second, in constructing the multi-head mechanism, the matrices of basis vectors $(\boldsymbol{e}_1, \boldsymbol{e}_2, \ldots, \boldsymbol{e}_n)$ can be shared.

## C  Corollary 1

Under the same conditions as in Theorem 3.1 and the value of $N$ is equal to the value of $d$, the Single-block attention representation Eq. 5 (in the paper) can reconstruct the *Scaled Dot-Product attention* in Eq. 2 (in the paper) by the summing over the tensor (i.e., the output of Single-block attention function) according to the second index. It holds that:

$$Attention(Q, K, V)_{i,m} = \sum_{j=1}^{N} Atten_{TD}(\mathcal{G}; Q, K, V)_{i,j,m}, \quad (7)$$

where $i$, $j$ and $m$ are the indices of the Single-block attention output (i.e., a 3-order tensor). $Atten_{TD}(\cdot)$ is the function of the Single-block attention based on Tucker decomposition. $i$ and $m$ are the indices of outputs (i.e., a matrix) from Eq. 2 (in the paper).

*Proof.* In Theorem B, we have proved the results about the attention function can be represented by a linear combination of basis vectors. Therefore, we can represent the self-attention function in Eq. 2 (in the paper) by the form as follows:

$$Attention(Q, K, V) = \Theta QK^T V \quad (8)$$

where $\Theta$ is a normalization factor matrix, which can be used to replace the use of a $sofmax$ function. We assume that $\Theta$ contains all the non-zero elements of the core tensor $\mathcal{G}$. The self-attention in Eq. 2 (in the paper) can be re-written as follows:

$$X_{i,m} = \sum_{k=1}^{N} \sum_{r=1}^{R} \Theta_{i,m} Q_{i,r} K_{k,r} V_{k,m} \quad (9)$$

where $N$ is the length of a sentence, $X_{i,m} = Attention(Q, K, V)_{i,m}$ is the entry of the output from the self-attention, and $R$ is equal to $d$. Here the core tensor $\mathcal{G}$ is same as that in Eq. 7 (in the paper). Then, the Single-block attention (a 3-order tensor) can be represented as follows:

$$\mathcal{A}_{i,j,m} = \sum_{p}^{R} \sum_{q}^{R} \sum_{r}^{R} \mathcal{G}_{p,q,r} Q_{i,p} K_{j,p} V_{m,r} \quad (10)$$

Figure 1: Tensor $\mathcal{A}$ is a 3-order tensor, which represents the Single-block attention in the left. $\mathcal{A}_{i,j,k}$ is the entry of the tensor $\mathcal{A}$. In the right, the graph represents that the summing of tensor slices which is from the tensor splitting in index $j$. This graph can help us to understand the main content of corollary C.

Figure 2: A diagram about a comparison of parameters between multi-linear attention and multi-head attention.

where $\mathcal{A}$ is a 3-order tensor, which is equal to $Atten_{TD}(\mathcal{G}; Q, K, V)$. Accordingly, $\mathcal{A}_{i,j,m}$ is a entry in tensor $\mathcal{A}$ and is equal to $Attention_{TD\,i,j,m}$ in Eq. 7. Next, we aim to prove Eq. 7 can be established. Therefore, we need to establish the relation between Eq. 10 and Eq. 9. Since the core tensor $\mathcal{G}$ is a special tensor (i.e., diagonal tensor), Eq. 10 can be written as follows:

$$\mathcal{A}_{i,j,m} = \sum_{r=1}^{R} \mathcal{G}_{r,r,r} Q_{i,r} K_{j,r} V_{m,r} \tag{11}$$

After that, we can compute the attention representation through adding to model $k$. For better understanding, we give the graph representation in Figure 1.

$$X_{i,m} = \sum_{j=1}^{N} \sum_{r=1}^{R} \mathcal{G}_{rrr} Q_{i,r} K_{j,r} V_{m,r}$$

The corollary then holds. $\qquad\square$

## D   Compression Ratio about Multi-Linear Attention

In order to compute the compression ratio, we need to compare multi-linear attention with multi-head attention. The comparison chart has been given in Figure 2.

In Figure 2, each $Linear$ function in multi-head attention is about a weight matrix $W \in \mathbb{R}^{d_{model} \times d}$, and all weight matrices in multi-head attention are different. In multi-linear attention, three weight matrices are used and $h$ (a number) weight vectors are used. Through the analysis about Figure 2, the

compression ratio is computed as follows.

$$\begin{aligned}
compression\ ratio =& \frac{3 \times h \times d_{model} \times d}{3 \times d_{model} \times d + h \times d} \\
=& \frac{3 \times h \times d_{model}}{3 \times d_{model} + h}
\end{aligned} \quad (12)$$

In practice, $h$ is equal to $8$ and $d$ is equal to $512$. The compression ratio approximates $8$ in this case. In our work, the dimension of vector $\mathcal{G}_r$ is set as $R$ which is smaller than $d$, where $d$ is the dimension of attention matrix.

**Low-rank Approximation for Model Compression** In the paper, we have described that our method combines two compression ideas, namely low-rank approximation and parameters sharing. Parameters sharing can be understood through the description of Figure 2. In Multi-linear attention, the idea of low-rank decomposition also has the function of model compression. We have proved that the Single-block attention can re-construct an one-head self-attention in Transformer. In order to obtain the representation of a tensorized attention, we adopt the tensor splitting and the concat function. After that, we consider that each tensor slice from tensor splitting approximates the output of the self-attention function Eq. 2 (in the paper). When we only focus on the idea of low-rank approximation, the compression ratio can be computed by the form, $\frac{N \times d}{N \times N}$, where $N$ is the length of a sequence, $d$ is the dimension of a matrix (also namely hidden size). $N$ is smaller than $d$, normally.

Through combining the ideas of parameters sharing and low-rank approximation, by formally considering the rank $R$, the compression ratio of Multi-linear attention model can be computed as follows:

$$compression\ ratio_R = \frac{3 \times h \times d_{model} \times d}{3 \times d_{model} \times d + R \times h}, \quad (13)$$

where $R$ is the rank of the core tensor $\mathcal{G}$. The compression ratio will be larger when $R$ is smaller. This $compression\ ratio_R$ is the compression ratio associated with $R$. $R$ need to be set in practice. In experiments, $R$ can be set to $18$, which is smaller than $d_{model}$.

# E    Experiment

## E.1    Partial Structure about Tensorized Transformer

in the paper, the multi-linear attention is proposed. In order to show that the process of incorporating multi-linear attention into Transformer, Figure 3 gives out some information about the structure.

## E.2    Experimental Details in Language Modeling

Now, we report some details of experiments as a relevant supplementary material. Firstly, we use three weight matrices $W^q, W^k$ and $W^v$ to linearly project the queries, keys and values. The outputs from the linear projections can be shared by $h$ times, where $h$ is the number of core tensors in our background (i.e., core-1($h$=1), core-2($h$=2)). We use Block Term Tensor decomposition (BTD) to construct a new representation, namely Multi-linear attention, which is a 3-order tensor. For incorporating the proposed attention into the architecture of Transformer, we split the 3-order tensor, and then concat each matrix from the tensor. For other layers, we use the same structure as vanilla-Transformer.

**Hardware**

We trained our model on one machine with 2 NVIDIA P40 GPUs. For our base models, the hyperparameters are described in Table 1. In addition, we set the $dropout$=0.3 in all datasets. The model is trained using 30 epochs in three datasets (PTB, WikiText-103 and One-Billion).

**Optimizer** We used the Adam optimizer and vary the learning rate over the course of training. The vary formula [8] is followed in our work. We also used the $warmup\_steps = 4000$. **Label Smoothing** is employed with the value $\epsilon$=0.1.

Figure 3: A diagram which is about the incorporating of multi-linear attention in partial Transformer structure. The parameters are shared in the constructing of each single-block attention.

Table 1: The hyperparameters in the Tensorized Transformers model

| Datasets | $d_{head}$ | $d_{ff}$ | h | L | $d_k$ | $d_v$ | Test PPL |
|---|---|---|---|---|---|---|---|
| PTB | 256 | 2100 | 2 | 3 | 40 | 40 | 49.8 |
| WikiText-103 | 256 | 2100 | 2 | 6 | 40 | 40 | 18.9 |
| One-Billion | 1024 | 2100 | 2 | 6 | 40 | 40 | 19.5 |

## E.3 Experiment Details in Neural Machine Translation

The Tensorized Transformer also has been applied to Neural Machine Translation task. In this experiment, we use the same setup with Transformer [8], and replace the multi-head attention with the proposed multi-linear attention in the encoder structure. In the decoder structure, we still use the multi-head attention for verifying the effectiveness of encoding a sentence. The model is trained in 1 NVIDA P40 GPUs.

## E.4 Experimental comparison

For a more detailed comparison, we design these experiments as follows. In this section, we mainly show the experimental results on two language modeling datasets, i.e., PTB and WikiText-103. We show the value of perplexity, as well as FLOPs[2] correspondingly.

| Model | PTB | | | WikiText-103 | | |
|---|---|---|---|---|---|---|
| | Params | FLOPS | Test PPL | Params | FLOPS | Test PPL |
| Transformer-XL [3] | 24M | 11.5B | 59.1 | 257M | 996.5B | 18.3 |
| Tensorized Transformer | 24M | 5.4B | 52.7 | 257M | 312.0B | 21.2 |
| Transformer-XL [3] | – | – | – | 151M | 126.5B | 24.0 |
| Tensorized Transformer | – | – | – | 151M | 83.4B | 18.8 |
| Transformer-XL [3] | 12M | 4.5B | 87.8 | 85.5M | 22.0B | 34.8 |
| Tensorized Transformer | 12M | 0.75B | 57.9 | 85.5M | 17.3B | 20.9 |

Table 2: Experimental comparisons on PTB and WikiText-103.

[2]FLOPs:The number of floating-point operations

Table 3: The same hyperparameters in Tensorized Transformers and Transformer-XL, $N$ is the length of sequence, and $L$ is the number of layers.

| Datasets | Model | $d_{head}$ | $d_{model}$ | $N$ | $L$ | dropout | Test PPL |
|----------|-------|-----------|------------|-----|-----|---------|----------|
| PTB | Transformer-XL | 40 | 256 | 30 | 3 | 0.3 | 81.2 |
| PTB | Tensorized Transformer | 40 | 256 | 30 | 3 | 0.3 | 50.2 |
| WikiText-103 | Transformer-XL | 40 | 256 | 80 | 6 | 0.1 | 34.86 |
| WikiText-103 | Tensorized Transformer | 40 | 256 | 80 | 6 | 0.1 | 19.9 |
| One-Billion | Transformer-XL | 40 | 1024 | 100 | 6 | 0.1 | 43.6 |
| One-Billion | Tensorized Transformer | 40 | 1024 | 100 | 6 | 0.1 | 26.7 |

To further compare the experimental results under the same size of parameters between Tensorized Transformer and the baseline model (i.e., Transformer-XL), we add some experiments in Table 2. In our paper, we use the Tensorized Transformer of 12M on PTB dataset and the Tensorized Transformer of 85.5M on WikiText-103 dataset. To achieve the same size, i.e., 12M for Transformer-XL on PTB, we can reduce the dimensions of $Q, K, V$ from 40 to 26. To achieve Transformer-XL of 85.5M, we reduce the dimensions of word embedding from 512 to 256. The experimental results are shown in Table 2. Our model gets better results and the lower FLOPS than Transformer-XL.

On the other hand, we can also increase the parameters of Tensorized Transformer to reach the parameters reported by Transformer-XL on PTB and WikiText-103 datasets. On PTB dataset, we can increase the number of layers from 3 to 7 to get the Tensorized Transformer of 24M. On WikiText-103 dataset, we increase the number of layers from 6 to 11 and the length of sequence from 80 to 120 to get the Tensorized Transformer of 257M. We can increase the number of layers from 6 to 8 and the length of sequence from 80 to 100 to get the Tensorized Transformer of 151M. After that, Tensorized Transformer achieves better results and lower FLOPS than Transformer-XL. These results are shown in Table 2.

In addition, we also carry out experiments when Transformer-XL has the same hyperparameters with Tensorized Transformer. Experimental results are shown in Table 3. Table 3 shows that our model can get the better results than Transformer-XL. Besides, on two datasets (i.e., PTB and WiliText-103), we also try to train our model (Tensorized Transformer) using larger dimension of word embedding (i.e., $d_{model}$). If $d_{model}$ is larger than 256 on PTB dataset and larger than 512 on WikiText-103 dataset, the overfitting will occur. For the overfitting problem, we will investigate it in our future work.

## F  Partial Code

The project have been achieved by pytorch. In this section, we give the partial code which is about our methods, i.e., Sing-block attention and Multi-linear attention. First, the class of Single-block attention is given as follows.

```python
import torch
import torch.nn as nn
import torch.nn.init as init
import numpy as np

class SingleBlockAttention(nn.Module):
    '''Single block attention'''
    def __init__(self, Rank):
        super(SingleBlockAttention, self).__init__()
        self.softmax = nn.Softmax()
        self.R = Rank
    def forward(self, q, k, v, mb_size,d):
        self.core = nn.Parameter(torch.FloatTensor(np.random.rand(self.R)))
        N = v.size(1)
        self.core = self.softmax(self.R)
        core_tensor = torch.zeros(N,d,N).cuda()
        for i in range(self.R):
            cores_tensor[i][i][i] = self.core[i]
        full_matrixs = []
```

```python
    for i in range(mb_size):
        full_matrix_1 = torch.einsum('pqk, ip,jq,kr->ijr', [core_tensor, q[i],
            k[i], v[i]]).contiguous()
        full_matrixs.append(torch.sum(full_matrix_1, dim=1))
    output = torch.stack(full_matrixs).cuda().float()
    return output
```

Each Single block attention is a component of Multi-linear attention. Based on the Single block attention, the Multi-linear attention can be given as follows.

```python
class MultiLinearAttention(nn.Module):

    ''' MultiLinearAttention '''

    def __init__(self, h, Rank, d, dropout=0.1):
        super(MultiLinearAttention, self).__init__()
        self.n_head = h # h is equal to 2 in our model
        self.d_k = d
        self.d_v = d
        self.w_q = nn.Parameter(torch.FloatTensor(d_model, d_k))
        self.w_k = nn.Parameter(torch.FloatTensor(d_model, d_k))
        self.w_v = nn.Parameter(torch.FloatTensor(d_model, d_v))
        self.Tattention = SingleCoreAttention(Rank)
        self.layer_norm = LayerNormalization(Rank)
        self.proj = Linear(self.n_head*d, Rank)
        self.dropout = nn.Dropout(dropout)
        init.xavier_normal_(self.w_q)
        init.xavier_normal_(self.w_k)
        init.xavier_normal_(self.w_v)

    def forward(self, q, k, v):

        d_k, d_v = self.d_k, self.d_v
        n_head = self.n_head
        residual = q
        mb_size, len_q, d_model = q.size()
        mb_size, len_k, d_model = k.size()
        mb_size, len_v, d_model = v.size()
        q_s = q.repeat(1, 1).view(-1, d_model)
        k_s = k.repeat(1, 1).view(-1, d_model)
        v_s = v.repeat(1, 1).view(-1, d_model)
        if n_head > 1:
          output_1 = self.Tattention(q_s, k_s, v_s, mb_size,d_v)
          output_2 = self.Tattention(q_s, k_s, v_s, mb_size,d_v)
          output = (output_1+output_2)*0.5
        else:
          ouput = self.Tattention(q_s, k_s, v_s, mb_size,d_v)
        # project back to residual size
        outputs = self.proj(outputs)
        outputs = self.dropout(outputs)
        return self.layer_norm(outputs + residual)
```

## Footnotes

*Corresponding Author: Peng Zhang