[Reviews · NeurIPS 2019]

Reviewer 1



The paper proposes a tensorized version of the transformer to reduce the number of parameters in the neural network architecture. The idea is original, and the empirical results are convincing on language modeling as well as machine translation. The writing however is not very clear (e.g. para 2 in intro, section 2.2) - please have your paper proofread for language and grammar. The results however are quite significant and will help scale the transformer to larger problems. Comments: 1. In theorem 3.1, the relation of the basis vectors to Q, K, V is confusing in the current notation. Might be useful to clearly specify it. 2. Did you try using more than 2 cores or a bigger model with more layers (esp. since the original transformer was limited by GPU memory)? 3. The paper is missing discussion on some very related work: a) Generating Long Sequences with Sparse Transformers Rewon Child, Scott Gray, Alec Radford, Ilya Sutskever **Edit**: Thanks to the authors for their response. Please add in the discussion of the sparse transformer in the related work and important details such as your definition of basis vectors to the main paper to improve the clarity.

Reviewer 2



The authors propose a new attention mechanism where the large amount of parameters that are required compute the affinities between the states and input units can be alleviated by sharing the parameters used to encode the different align-able units. As such, the number of parameters needed to incorporate the attention mechanism reduces substantially, and additionally, results on language modelling and translation seem to indicate that the proposed mechanism generalizes better with fewer parameters. The paper provides a good self-contained extension the transformer class of models, which is worth publishing.

Reviewer 3



# Clarity I had difficulty understanding how the authors implemented their solution, so I looked at the code in the supplementary material. This code failed to compile, and had numerous confusing aspects, and the authors did not link to the actual code used in training the model. *Confusing aspect of code #1* There are some parts of the code that are not valid python. For example: ``` for i in range(): cores_tensor[i][i][i] = self.core[i] ``` *Confusing aspect of code #2* Within the MultiLinearAttention class, the authors provide the following branching logic: ``` if n_head > 1: output_1 = self.Tattention(q_s, k_s, v_s, mb_size,d_v) output_2 = self.Tattention(q_s, k_s, v_s, mb_size,d_v) output = (output_1+output_2)*0.5 else: ouput = self.Tattention(q_s, k_s, v_s, mb_size,d_v) ``` My understanding is that `self.Tattention` is a pure function, therefore the True branch should always be numerically identical to the False branch, but at 2x the cost. === # Originality This result seems motivated by the same problem as "Generating Long Sequences With Sparse Transformers" (https://arxiv.org/abs/1904.10509 April 2019), but I could find no comparison with that work. === # Significance I think that the results of using 50% fewer parameters is quite significant. However I would also like to see the total flops usage compared to the baseline, as flops are frequently the limiting factor for training and deployment of models.

[Author Response · NeurIPS 2019]

We highly appreciate the reviewers' invaluable comments and suggestions. Our responses are as follows.

To **Reviewer 1**: For Theorem 3.1, Q, K, and V can be linearly represented by a set of basis vectors. Such a relation
is detailed in the supplementary material (in Eq.2). Under this condition, we prove Eq.4 in our paper. Theorem 3.1
provides a theoretical basis for building the proposed Multi-linear attention.

We used 3 or 4 cores to train our model. In the Future Work section of the paper, we mentioned that our model may
suffer from overfitting when the number of cores becomes larger, and the issue will be explored in future. Furthermore,
on PTB dataset, we have tested our model when increasing the number of layers from 3 to 7, and our model has
approximately equal parameters (24M) as the original Transformer. However, our model gets a lower perplexity (PPL
52.7) than the original Transformer (PPL 59.1). We will add a more systematic evaluation in the revised version.

A detailed discussion about Sparse Transformer is given in the end of this response.

To **Reviewer 2**: According to your suggestion, during the response period we have tested the Transformer with reduced
parameters by directly reducing the dimensions of Q, K, and V from 40 to 26. We have run the language modeling
experiments on PTB dataset. This method achieves the same parameters (12M) as in our tensorized model, while
obtaining a PPL 87.8, which is higher than the PPL 57.9 of our model. The result shows the better performance of our
model. We will report a systematic comparison results in the revised paper.

Regarding the rationale for the improvements, besides the alleviation of overfitting by reducing parameters, another
reason is that our method captures more information than the original Transformer. In Corollary 1, we prove that the
output of the original attention can be represented by summing over the 3-order tensor (i.e., $output(\text{original})=\sum_i^n T_i$,
where $T_i$ is matrix resulting from splitting the 3-order tensor(N×N×N), in Figure 2 of our paper). In our tensorized
transformer, we use a concat function over these matrices ($ouput(\text{tensorized})=concat(T_1,\ldots,T_i)$). The operation of
concat models all values in the 3-tensor, and thus captures more information than the operation of sum.

To **Reviewer 3**: (1) For grammar issues, we will carefully check them and make changes accordingly.
(2) For the code link you mentioned, since the Author Response policy of NeurIPS2019 explicitly states "Author
Responses must not contain external links", we can not provide such a link. Instead, we clarify the confusing aspects
you mentioned and explain how to compile it as follows.

| for i in range():    (a)<br>   cores_tensor[i][i][i] = self.core[i] | self.T_attention=    (b)<br>SingleCoreAttention(Rank) | "  self.core =nn.Parameter(torch.FloatTensor(np.random.rand(self.Rank))) (c)<br>def forward(self, q, k, v, mb_size, d):" |
|---|---|---|
| for i in range(self.R):  ↓<br>   cores_tensor[i][i][i] = self.core[i] | self.Tattention=  ↓<br>SingleBlockAttention(Rank) | "def forward(self, q, k, v, mb_size, d):  ↓<br>   self.core =nn.Parameter(torch.FloatTensor(np.random.rand(self.R)))" |

Figure 1: Code correction: the code in first line should be corrected to the code in second line
(3) We apologize for the clerical errors we made when editing the code with Latex. We have corrected them in Figure 1.
#Confusing aspect of code 1: missing the variable "self.R" in parentheses. The revision is in Figure 1(a).
#Confusing aspect of code 2: the "self.Tattention" is not a pure function. It is the class named "SingleBlockAttention".
The revision is in Figure 1(b). In addition, the True branch is not numerically identical to the False branch. In the class
of "SingleBlockAttention", we define a variable "self.core" using a random function. The position for the definition of
"self.core" should be adjusted. The revision is in Figure 1(c).

The code in supplementary file is the key code that aims to help readers understand the Multi-linear attention. This
code can be compiled through the following steps. a) "SingleBlockAttention" replaces "ScaleDotProductAttention" in
Transformer , and "MultiLinearAttention" replaces "MultiHeadAttention" in Transformer; b) Setting the parameter
"head" in Transformer to 1. c) Changing the dimensions of the matrix in the fully connected layer.

We will add the experimental results for the total flops usage in comparison with the baseline in the next version.
We will add the following discussion on Sparse Transformer to the Related Work section in the revised version. We will
also include the experimental results of Sparse Transformer in the language modeling task in comparison with ours.

**Discussion of Sparse Transformer**: Sparse Transformer is a recent work that adopts sparse techniques on the attention
matrix and reduces its parameters. However, it differs from our Tensorized Transformer in the following aspects.
(1) Sparse Transformer does not change the computation of original attention, but uses a sparse attention matrix by
selecting the information on some positions in the attention matrix. Our paper proposes a new attention structure named
Multi-linear attention. (2) Sparse Transformer does not outperform the original Transformer, as its sparse attention
matrix only contains partial information of the original one. The Multi-linear attention we propose is based on tensor
decomposition. It not only reduces parameters, but also improves the results. The improvements come from the use of
the concat function in Multi-linear attention which captures more information. Please find detailed explanation in the
response "To reviewer 2". (3) The target tasks in Sparse Transformer often involve processing of images, text (character
predict) and audio. The target tasks of our model are language modeling and neural machine translation. When dealing
with text, Sparse Transformer is trained at the character level, while language modeling task in our paper is typically on
the word level.

[Meta-Review · NeurIPS 2019]

This paper reformulates the attention component of the classic Transformer language model in terms of tensor operations. The reviewers agree that the proposed model is well motivated and that the reduction in parameters achieved is significant. As such this paper is worthy of publication. However the reviewers also note a number of issues with the clarity of the presentation, general grammatical errors, and errors in the accompanying code. All of these issues must be addressed before publication. It is also required to add a more complete evaluation across a range of parameters scales for the tensorised model and the baseline, and to include the total flops used.